# Factors determining antibiotic use in the general population: A qualitative study in Spain

Olalla Vazquez-Cancela[1,2], Laura Souto-Lopez[1], Juan M. Vazquez-Lago[1,2,3]*, Ana Lopez[4], Adolfo Figueiras[1,3,5]

**1** Department of Preventive Medicine and Public Health, University of Santiago de Compostela, Santiago de Compostela, A Coruña (Corunna), Spain, **2** University Hospital of Santiago de Compostela, Santiago de Compostela, A Coruña (Corunna), Spain, **3** Health Research Institute of Santiago de Compostela (IDIS), Santiago de Compostela, Spain, **4** Department of Clinical Psychology and Psychobiology, University of Santiago de Compostela, Santiago de Compostela, A Coruña, Spain, **5** Consortium for Biomedical Research in Epidemiology & Public Health (*CIBER en Epidemiología y Salud Pública - CIBERESP*), Spain

* juan.manuel.vazquez.lago@sergas.es

**Data Availability Statement:** All relevant data are within the manuscript and its Supporting information files.

## Abstract

### Background

Antibiotic resistance is an important Public Health problem and many studies link it to antibiotic misuse. The population plays a key role in such misuse.

### Objective

The aim of this study was thus to explore the factors that might influence antibiotic use and resistance in the general population.

### Methods

Qualitative research using the focus group (FG) method. Groups were formed by reference to the following criteria: age (over and under 65 years); place of origin; and educational/professional qualifications. FG sessions were recorded, transcribed and then separately analysed by two researchers working independently. Written informed consent was obtained from all participants.

### Results

Eleven FGs were formed with a total of 75 participants. The principal factors identified as possible determinants of antibiotic misuse were: (i) lack of knowledge about antibiotics; (ii) doctor-patient relationship problems; (iii) problems of adherence; and, (iv) use without medical prescription. Antibiotic resistance is a phenomenon unknown to the population and is perceived as an individual problem, with the term "resistance" being confused with "tolerance". None of the groups reported that information about resistance had been disseminated by the health care sector.

**Funding:** This work was supported in part by the Instituto de Salud Carlos III (PI081239, PI09/90609) Spanish State Plan for Scientific and Technical Research and Innovation 2012-2016, The European Regional Development Fund (ERDF). https://www.isciii.es/QueHacemos/Financiacion/Paginas/default.aspx.

**Competing interests:** The authors have declared that no competing interests exist.

## Conclusions

The public is unaware of the important role it plays in the advance of antimicrobial resistance. There is evidence of diverse factors, many of them modifiable, which might account for antibiotic misuse. Better understanding these factors could be useful in drawing up specific strategies aimed at improving antibiotic use.

## Introduction

Taken together, antibiotic adverse effects, ineffectiveness and resistance is one of the biggest threats to global health [1], due to the great impact on morbidity, mortality and costs [2]. Over- and misuse of antibiotics contributes significantly to this problem [3]. Indeed, overuse must be assumed to account for the differences in antibiotic use (as much as threefold) among European Union countries [4], due there is no evidence of any difference in the prevalence of infectious diseases [5].

Most antibiotic use (80% to 90%) occurs in the outpatient setting [6, 7]. In terms of antibiotic consumption, Spain not only ranks highest among developed countries (in excess of 40 Define Daily Dose (DDDs) per 1,000 inhabitants per year), but its figures continue to rise [8]. Furthermore, around 30% of all outpatient antimicrobial sales are not identified from reimbursement data, due in large part to the existence of non-prescription sales [9, 10]. While physicians, pharmacists and health authorities are all involved in antibiotic over- and misuse, patients may also play an important role, in that: (i) they are the end-users and can decide whether or not to take antibiotics or to suspend the treatment; (ii) they can demand antibiotics at the pharmacy without medical prescription; and, (iii) they can exert pressure on physicians to prescribe or on pharmacists to dispense these antibiotics [9, 11–13].

Despite the key role that the public may play in the advance of resistance, the factors that influence antibiotic misuse in the general population remain unknown [14], something that hinders the design of specific purpose-designed strategies [15]. Accordingly, the aim of this study was to use qualitative methodology to examine factors identified by the public as being responsible for antibiotic use and misuse.

## Methods

### Study design

The study was undertaken in Galicia, a region in north-west Spain which has a population of 2.7 million [16] and registers high levels of antibiotic use, with a figure of as much as 23 DDD per 1,000 inhabitants per year recorded in 2016 [17].

A qualitative study was conducted, using the focus group (FG) technique as a tool for collecting narrative data. The choice of qualitative methodology was determined by the fact that it allows for in-depth examination of population attitudes to antibiotic use: the FG technique is the best tool for generating interactive discussion and addressing subjective aspects from diverse points of view, something that is difficult to achieve with quantitative methods [18, 19].

### Selection, sample and procedure

We sought to ensure a high degree of heterogeneity in the composition of the groups in terms of age (over and under 65 years), urban or rural origin, and educational/professional qualifications, in order to cover the widest range of opinions (Table 1). We made groups following age

**Table 1. Focal group characteristics.**

| FG aged >65 years | n | M:W | Population | Professional healthcare qualifications | FG aged <65 years | n | M:W | Age participants | Population | Professional healthcare qualifications |
|---|---|---|---|---|---|---|---|---|---|---|
| FG1 | 6 | 1:5 | Rural | - | FG6 | 5 | 0:5 | >50 | Urban | 1 Pharmacist |
| FG2 | 5 | 2:3 | Urban | - | FG7 | 5 | 1:4 | >50 | Rural | 1 Biologist |
| FG3 | 9 | 2:7 | Urban | - | FG8 | 6 | 3:3 | <35 | Urban | - |
| FG4 | 8 | 0:8 | Urban | - | FG9 | 5 | 2:3 | >50 | Rural | 1 Nurse |
| FG5 | 8 | 2:6 | Rural | - | FG10 | 12 | 3:9 | 35–50 | Urban | - |
|  |  |  |  |  | FG11 | 6 | 3:3 | <35 | Urban | 1 Biologist |

M: Men

W: Women

criteria to explore the differences in knowledge and attitudes between retirees and workers. We decided to made this two groups to better explore the differences in the acces to the heathcare facilities (assuming more time in retirees), and also to explore the differences in the relationship with the doctor between older and yougers. We also took into account the origin criteria due to possible differences in access to the health system. The help of key informants and the snowball method were used [20]. The heads of 50 socio-cultural associations, senior citizen study centres and neighbourhood associations were contacted by e-mail and telephone. At a meeting held with the 16 centres that responded to our invitation, we explained what the study consisted of and its aims. Of the original sixteen centres, three refused to participate, one due to a lack of interest and the other two due to an insufficient number of members. In addition, a further two groups were ruled out because saturation of information had been achieved with 11 FGs. As a result, no new group sessions were convened [21].

We drew up a script so as to conduct the sessions in line with the conclusions of previous studies on general practitioners (GPs) [12, 22] and community pharmacists [23], with the ultimate aim of testing these findings on and with the help of the public. In addition, we conducted a bibliographic review of papers published on the subject to date [14, 24–33], requesting the authors for their respective scripts so as to include all relevant topics [28, 31–33]. Expert researchers in qualitative methodology (ALD, AFG, JMVL) collaborated in drawing up the script, to ensure open-ended questions and a permissive environment conducive to the free flow of the participants' discourse and the veracity of the opinions voiced.

The FGs were guided by two researchers (OVC, LSL). At the end of every session, a summary was drawn up detailing the group's characteristics and first impressions.

A digital audio recorder was used. The sessions had a duration of approximately 45 minutes each, and came to an end when no more new ideas or contributions were forthcoming from the participants. An informal training session on antibiotic use was offered at the end and 4 groups requested this, with the result that their sessions were extended for an extra 40 minutes. One researcher made the literal transcriptions, endeavouring in every case to take no longer than 5 days after the session, and a second observer was responsible for checking and correcting any possible errors on the basis of consensus. Participants were coded by range age and gender ("M" for men, "W" for women), and each group was identified with a serial number (FG1, FG2, FG3, etc.).

## Ethical considerations

The study was evaluated and approved by the Santiago-Lugo Research Ethics Committee. After being informed of the purpose of the study and the fact that the sessions were to be

recorded and transcribed but kept anonymous, all the participants agreed to take part and gave their written informed consent.

## Analysis

The transcriptions were analysed separately by two researchers (LSL, OVC), in the interests of reducing any risk of researcher bias.

A thematic and discourse analysis of the data was performed, and was then discussed by all the authors. Ideas were identified, and the data obtained were organised by topic area and accompanied by literal excerpts, which served as units of analysis. Subsequently, the ideas extracted were associated with pre-established variables using the grounded theory method [34]. Any disagreements as regards interpretation were discussed by the researchers and resolved by consensus. No computer software programme was used for processing the data.

## Results

In the period from March to May 2017, eleven FGs, each containing 5 to 12 members, were formed, making a grand total of 75 participants (Table 1).

After analysis of the recordings, the main reasons given by the public to explain antibiotic misuse and abuse (Table 2) were identified as being: (i) lack of knowledge about antibiotics; (ii) problems in the doctor-patient relationship; (iii) problems of adherence; and, (iv) use without prescription. Additionally, the following were also identified, even though they were not cited as reasons *per se*: (v) lack of perception of the problem; and, (vi) external attribution of responsibility (Table 3).

**Table 2. Coding of the results identified in the population.**

| | |
|---|---|
| **Lack of knowledge about antibiotics** | • Difficulties in differentiating antibiotics from other medications.<br>• Consider that antibiotics are used for any infection. |
| **Problems in the doctor-patient relationship** | • Lack of trust in physician (pressure on physician).<br>• Consider that the physician supplies little information about the disease.<br>• Consider time of consultation to be insufficient. |
| **Problems of adherence (not finishing the entire treatment)- Reasons** | • Lack of credibility of professional judgement<br>• Improvement after initial doses<br>• Side effects of antibiotics<br>• Abandoning the treatment in order to be able to consume alcohol<br>• Oversights, carelessness |
| **Use without prescription** | • Trusted pharmacy<br>• Home medicine cabinet/leftover antibiotics<br>• Internet |
| **Lack of perception of the problem of development of resistance** | • Do not think that there is any problem at present<br>• Excess use of antibiotics is not linked to advance of resistance<br>• Not considered to be a Public Health problem |
| **Responsibility** | • Internal: inappropriate use of antibiotics considered responsible for the problem.<br>• External (considering other being responsible of the problem): physicians, pharmaceutical industry, food, economic reasons, excess use in the past considered responsible for the problem. |

**Table 3. Results of the FG sessions.**

| FG1 | FG2 | FG3 | FG4 | FG5 | Factor | FG6 | FG7 | FG8 | FG9 | FG10 | FG11 |
|---|---|---|---|---|---|---|---|---|---|---|---|
| X | X | X | X | X | **Problems of knowledge** | X | X | X | X | X | X |
| | | | X | X | **Doctor-patient relationship problems** | X | X | X | X | X | X |
| X | X | X | | X | **Problems of adherence** | X | X | X | X | X | X |
| X | X | X | X | X | **Use without prescription** | X | X | X | X | X | X |
| X | X | X | X | X | **Lack of perception of the problem of development of resistance** | X* | X* | X | X* | X | X* |
| X | | | | X | **Internal responsibility** | | | | X | | |
| X | X | X | X | X | **External responsibility** | X | X | X | X | X | X |

*In these groups, one person understood the magnitude of the problem as a result of holding specific healthcare qualifications, as shown in Table 1

## Lack of knowledge

In all the over 65 age FGs, at least one participant in each group was unable to differentiate between antibiotics and other types of medication, either asking for clarification or displaying indiscriminate use of the terms while speaking.

While the under 65 age FGs were clear as to the difference, at least one participant in each group was ignorant of the fact that antibiotics were ineffective in the case of viral infections.

Lack of knowledge was considered to be one of the factors of misuse: ["*People don't realise that antibiotics don't combat viruses, and most infections are viral, but they take antibiotics because they don't know how to use them*"] (>65y, M6, FG1).

This lack of knowledge means that antibiotics are mistakenly regarded as faster-acting and more efficacious medications: ["*Don't give me just any old remedy, give me one that'll cure me, give me an antibiotic*"] (>65y, W2, FG1); ["*When I have a cold, of course I'd like to take an antibiotic; I feel really bad and I want an antibiotic, obviously because I think that way I'll get rid of it more quickly*"] (51-65y, W2, FG6).

Fever was reiterated by four over-65 FGs and one under-65 age FG as one of the symptoms that requires antibiotics: ["*But if you've got a temperature, and you go to the doctor, what's he going to give you unless it's an antibiotic?*"] (>65y, W1, FG2).

Only two groups saw the medical practitioner as being responsible for taking the decision to prescribe antibiotics, once the necessary check-up and examination had been performed: ["I think it is necessary a severe control in the antibiotics. Doctors are the ones who always have to make the decision (taking or not antibiotics)"] (>65y W6, FG1). Other groups stated that in some illness any person can know that you need an antibiotic, even without a medical examination: ["Here with all the cold we have, you can get an urine infection. A simple urine infeccion, and you don't have more remedy than take an antibiotic."] (>65y, W4, FG5).

## Poor doctor-patient relationship

Poor doctor-patient relationship was highlighted, especially in the under 65 age group: ["*I think that doctors need to learn how to talk to patients. The way they speak to and handle patients, that's what's got to improve*"] (51-65y, W5, FG6). Participants complained of the lack of information and explanations given by physicians: ["*Doctors tend to be pretty evasive and tell you very little . . .it's not good to rush things*"] (51-65y, M1, FG7).

It was felt that a poor relationship can affect trust, and thus lead to a weakening of medical judgement. This was associated with the pressure which patients put on physicians to prescribe antibiotics: ["*People ask for medicine because their GP is the kind of doctor who's heard it all before, so the patient wants to make sure she's going to improve, since she believes that it's only*

*with antibiotics that she'll be able to get better, because she doesn't understand, seeing as they don't tell her what she's got*"] (51-65y, W5, FG6).

Lack of credibility in the health professional translates as a search for alternatives, such as going to the emergency ward or seeking a second opinion from a private physician: ["*If your GP doesn't given you them (antibiotics), well you go to emergencies: if you're convinced that you really need them, I think you'll get them in the end*"] (18-34y, M2, FG8) ["*There are people who go to the GP in the morning and the GP doesn't give them any (antibiotics)... in the afternoon they go to emergencies, so that they'll give them some. Or you go to a private doctor and they'll also give them to you*"] (51-65y, W2, FG9).

## Problems of adherence (not finishing the entire treatment)

In all groups but one (FG4), the participants disclosed problems of adherence. The reasons for abandoning treatment were improvement after initial doses, fear of side effects ["*90% of the times in my life that I've taken antibiotics for an infection I've ended up getting ill from something else... or my stomach or whatever...*"] (18-34y, W2, FG11), oversights, and specific abandonment of treatment so as to be able to consume alcohol (FG2, 10).

Loss of credibility and trust in the physician were identified as important reasons for lack of adherence to the prescribed treatment: ["*I think that, if we patients more or less followed the doctor's instructions and those that come with the medicine, I mean to say there's a lack of trust*"] (18-34y, W3, FG8).

Despite the fact that problems of adherence were identified in all groups, doubts about the treatment guideline as prescribed by the physician was not cited as a reason for misuse: ["*Sometimes they give you a note and tell you how you have to take it. They put 'two a day', or 'three a day'...*"] (>65y, W1, FG3). Two groups pinpointed the pharmacy as the place where doubts were resolved ["*Very often, pharmacies are the ones that help you clear things up*"] (51-65y, W1, FG9).

## Antibiotic use without prescription

There was acknowledged use without prescription, whether by going to trusted pharmacies or by using leftover antibiotics from previous illnesses (home medicine cabinet), associated with people's belief in their ability to recognise situations in which antibiotics are required: ["*I think they self-medicate because they had -or think they had- the same illness, and they still have some drugs left over from last time*"] (18-34y, M1, FG8).

Eight groups admitted to having a home medicine cabinet and resorting to it when they thought it was necessary: ["*We don't throw anything anyway; who doesn't have a medicine cabinet at home?*"] (>65y, M5, FG4). In eight groups, the idea of going to a trusted pharmacy to obtain antibiotics was raised ["*I go to the pharmacy and I say to him, what'll you give me? For urinary infections, they always gave it to me (...) at the pharmacy, provided it's one you trust, but to be honest, they wouldn't have given it (the antibiotic) to me, if they hadn't known me*"] (51-65y, W1, FG9). When it came to the difference between resorting to a home medicine cabinet and a trusted pharmacy, the former measure was perceived as negligent, whereas the latter was perceived as an appropriate alternative avenue.

No group reported difficulty of access to the health-care system. However, in six of the groups (4 of which were over 65 years old), people said that they avoided going to the physician and only went as a last resort ["*I'm not one for going to the doctor [...], I'm not at all keen. If it's strictly necessary, I'll go; if it isn't, I won't.*"] (>65y, W1, FG1).

### Lack of perception of the problem of antibiotic resistance

Antimicrobial resistance is regarded as a problem of individual consumption, with no distinction been drawn between resistance and tolerance: ["*I have a certain respect for antibiotics, because I don't want my body to get used to them, and then when I really need them. . . they don't work.*"] (18-34y, M4, FG11).

Although antibiotic resistance is of concern to the public, its advance is not associated with misuse. Difficulty in finding effective antibiotics is considered a consequence of intensive farming and food, insted of human misuse: ["*All the chickens that come from intensive farming, for example, have antibiotics; and the cattle. . .have them in the meat as well as in the milk. . .*"] (18-34y, M5, FG11).

Only the 4 participants with specific healthcare qualifications (2 biologists FG7 and FG11, 1 nurse FG9, and 1 pharmacist FG2) stated that they understood the magnitude of the problem. In these groups, lack of information was considered the principal problem: ["*Resistance is due to a lack of information, the public's profound lack of information and awareness*"] (51-65y, W1, FG9). Groups that displayed worse comprehension of the problem felt that they had sufficient knowledge: ["*I don't think there's any lack of information, hey! nowadays we're very well informed*"] (>65y, W3, FG4).

Information on antibiotic resistance did not come from health professionals in any of the groups but was instead obtained from the press and other mass media: ["*Whenever I go to the doctor, he gives me antibiotics, and that's all there's to it. Don't go telling me, be careful because the bug is getting stronger due to people like you taking antibiotics"]* (18-34y, W2, FG11). This information has been disseminated without attaining public health relevance: ["*I think it's the responsibility of each one of us; and what other people do is all the same to me*"] (18-34y, M4, FG11).

Physicians, the pharmaceutical industry and food are blamed for the advance of resistance. Yet, public education and awareness raising by the health care sector is nevertheless regarded as essential: ["*The health professional has to do a job of awareness-raising, if it not at a personal and human level, then at the level of a publicity campaign; people have to be made aware that taking an antibiotic is no idle matter*"] (35-50y, M10, FG10).

## Discussion

This is the first qualitative study undertaken in Spain to explore the factors that influence people in terms of their use of antibiotics and its relationship with resistance. Our study shows that the public is unaware of the important role it plays in the advance of antimicrobial resistance. It also highlights the fact that lack of knowledge and doctor-patient relationship problems influence antibiotic use. Knowledge of these factors will enable more specific strategies to be implemented, with the aim of improving antibiotic use and increasing the impact of awareness-raising campaigns [15].

Our study served to detect crucially important gaps in public knowledge, revealing that people: (1) do not understand the difference between viral and bacterial infections; (2) think that symptoms such as fever should be directly treated with antibiotics; (3) believe that excess use of antibiotics is unconnected with the advance of resistance (with industrial livestock farming and food processing being to blame); (4) cannot differentiate between tolerance and resistance; and (5) are unaware of the dimension of the public health risks posed by resistance. These gaps could be accounted for by the fact that previous awareness-raising *campaigns* have been based on informing the public about excess use and the consequences of not completing a course of treatment [15, 35]. Our study also indicates that the population is extremely receptive to more training in this field, something that could provide a good opportunity for well-designed interventions to be effective.

Our results show that the public demands antibiotics because it does not trust clinical diagnosis and, at the same time, does not complete the course of treatment for fear of side effects. Moreover, there was evidence to show that a poor *doctor-patient relationship and communication* makes for a loss of credibility in respect of medical advice and worse adherence to treatment [36]. Patients complained that neither the treatment nor the importance of their illness was explained to them [37], and consider more information and communication by health professionals to be necessary. To our study population's way of thinking, this justifies the *pressure* that they bring to bear on physicians when it comes to seeking treatments. Previous studies conducted in the same geographical setting found complacency to be one of the main motivations acknowledged by physicians and pharmacists alike, when it came to prescribing and dispensing antibiotics [12, 23]. This is in contrast to the view of patients, who consider that physicians should not succumb to pressure, a finding that is consistent with other previous studies in which physicians were observed to overestimate patients' expectations [38, 39]. In contrast, dispensing without prescription was perceived in our study as something done as a favour by the pharmacist.

This poor *doctor-patient relationship and communication* is also associated with a lack of *credibility* in medical judgement, with the result that patients seek alternative ways of obtaining antibiotics: (1) they admit to making use of the emergency services to get prescriptions for antibiotics in situations where, faced with a refusal on the part of their GP, they nonetheless regard them as necessary. This disparity in criteria between primary and emergency care may weaken the doctor-patient relationship still further. To prevent this, antibiotic optimisation programmes should be extended to the emergency services, using the same criteria as in primary care [39–41]. (2) Another alternative is to resort to the use of the *home medicine cabinet or their trusted pharmacy* to obtain antibiotics without prescription. This might go some way to account for the fact that 30% of antibiotic use takes place outside the health care system [10]. Whereas demand for antibiotics from health professionals is motivated by concern about and problems in the doctor-patient relationship, self-medication, on the other hand, is associated with the belief in the ability to recognise the disease by virtue of having suffered from similar symptoms previously. Our study population insisted that the pharmacies to which they resorted had to be trusted. These results were in contrast to pharmacists' belief that, if they did not relent, patients would obtain the antibiotics at some other pharmacy [9, 23].

In our study, the public did not report difficulties in *access to the health-care system* which would justify the search for alternatives to consulting a physician. Even so, they avoid going to the doctor, and when they do go, it is to receive treatment and not medical advice. This goes to show that the existence of a poor doctor-patient relationship is an important gap to be borne in mind.

## Strengths and limitations

**Limitations.** The FG sessions took place in Galicia, an area with a population that has a high use of antibiotics without prescription. Prudence is therefore called for when generalising the findings to other regions of Spain. It is necessary to replicate this workin other parts of Spain. Other natural limitation include the non-random sample, participants were volunteer. We don't see this as a big limitation because the participants represented a wide range of ages, origin and formation.

**Strengths.** Eleven FGs were formed, taking into account differences in age, origin (urban or rural) and educational/professional qualifications. The methodology and design used were in line with the quality criteria required by qualitative techniques. The study fulfilled all COREQ scale criteria [42], except for point 23 (Transcripts returned) which did not prove feasible, owing to the characteristics of the population, namely, an elderly age stratum, without

any available means for delivery of transcriptions. By way of correction, however, separate transcriptions were drawn up by two researchers, with any points of difference being discussed and settled by common agreement.

## Conclusions

Improving antibiotic use is a complex task that calls for a number of complementary approaches. One of the targets must be patients, due to their key role in the correct use of antibiotics. Qualitative population studies and a systematic review have both highlighted the importance of lack of knowledge. Our study goes further still and highlights the importance of the doctor-patient relationship and proper transmission of information to the patient, not only at the level of the individual consultation, but also at the level of public health campaigns. These findings may well be of great utility when it comes to designing more direct, higher-impact campaigns aimed at improving antibiotic use in and by the general population.

## Supporting information

**S1 Checklist. COREQ (COnsolidated criteria for REporting Qualitative research) Checklist.**
(PDF)

**S1 File.**
(DOCX)

## Acknowledgments

We should like to thank all the neighbourhood associations and senior citizen study centres that kindly collaborated in this study.

## Author Contributions

**Conceptualization:** Juan M. Vazquez-Lago, Ana Lopez, Adolfo Figueiras.

**Data curation:** Olalla Vazquez-Cancela, Laura Souto-Lopez.

**Formal analysis:** Olalla Vazquez-Cancela, Laura Souto-Lopez.

**Investigation:** Olalla Vazquez-Cancela, Laura Souto-Lopez.

**Methodology:** Juan M. Vazquez-Lago, Adolfo Figueiras.

**Project administration:** Juan M. Vazquez-Lago, Adolfo Figueiras.

**Supervision:** Juan M. Vazquez-Lago.

**Validation:** Juan M. Vazquez-Lago, Ana Lopez, Adolfo Figueiras.

**Writing – original draft:** Olalla Vazquez-Cancela, Laura Souto-Lopez.

**Writing – review & editing:** Juan M. Vazquez-Lago, Ana Lopez, Adolfo Figueiras.

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
