## [Decision Letter · Decision Letter 0]

25 Jun 2020

PONE-D-20-05804

Factors determining antibiotic use in the general population: a qualitative study in Spain.

PLOS ONE

Dear Dr. Vazquez-Lago,

Thank you for submitting your manuscript to PLOS ONE. After careful consideration, we feel that it has merit but does not fully meet PLOS ONE’s publication criteria as it currently stands. Therefore, we invite you to submit a revised version of the manuscript that addresses the points raised during the review process.

We look forward to receiving your revised manuscript.

Kind regards,

Helena Legido-Quigley, Ph.D.

Academic Editor

PLOS ONE

Journal Requirements:

2. Please include additional information regarding the interview guidelines used in the study and ensure that you have provided sufficient details that others could replicate the analyses. For instance, if you developed interview guidelines as part of this study and it is not under a copyright more restrictive than CC-BY, please include a copy, in both the original language and English, as Supporting Information.

3. Please ensure that you include a title page within your main document. We do appreciate that you have a title page document uploaded as a separate file, however, as per our author guidelines (http://journals.plos.org/plosone/s/submission-guidelines#loc-title-page) we do require this to be part of the manuscript file itself and not uploaded separately.

Reviewers' comments:

Reviewer's Responses to Questions

**Comments to the Author**

1. Is the manuscript technically sound, and do the data support the conclusions?

Reviewer #1: Yes

Reviewer #2: Partly

2. Has the statistical analysis been performed appropriately and rigorously? 

Reviewer #1: N/A

Reviewer #2: No

3. Have the authors made all data underlying the findings in their manuscript fully available?

Reviewer #1: Yes

Reviewer #2: No

4. Is the manuscript presented in an intelligible fashion and written in standard English?

Reviewer #1: No

Reviewer #2: No

5. Review Comments to the Author

Reviewer #1: General comments:

The topic is of great interest where antibiotics abuse and misuse is a serious public health concern.

The authors used adequate and solid methods for qualitative research.

Providing the scripts that the authors used to conduct the sessions as appendices facilitates the replication of the methods.

Manuscript would benefit from a thorough proof read by a native English speaker to improve grammar.

Specific comments:

Line 43: spell out DDD when used for the first time in text

Line 143: Better explain in the methods section the rationale for dividing FGs between over 65 and under 65

Line 163: authors should explain what they mean by M1 (same for lines 168, 171, 183…)

Line 176: authors should explain what they mean by H.1

Lines 356-357: Conclusions are usually free of citations especially if the articles are cited earlier in text; unless they mention a quote or an expert opinion.

Table 1: Providing more details about the baseline characteristics would give more insights to the respondents’ characteristics and would help understand the data better.

Reviewer #2: This is an important topic and authors made a clear important argument on the study purpose. However, there are important parts which require significant revision, also language requires revision and polishing as it is difficult to understand.

1) Introduction:

a)1st para: unclear what author tried to express. Line 35--( delete "it has on"); Line 38-39 please revise and double check the english grammar

b)Line 51 change these drugs to antibiotics

c)Line 57 abuse, can you define "abuse" and add this in the 2nd para

2)Methods

a) Line 72, the references 18,19 did not match in Line 68-72

b) Line 76-79, how the snowball method approved to be high degree of heterogeneity ?

c) Line 85, how did the authors know the "saturation of information had been achieved" without knowing what these two groups' discussion

d) Line 92: what was the response rates for requesting other authors

e) Ethical approval reference No ?

f) It was unclear if the discussion was conducted in Spanish or English ? transcript was translated into english ? how did the author make sure there was a back translation?

3) Results

a)Line 143 and Line 147 over-65? over 65 age, please double check the English grammar

b) add the definition of the summarised 5 main reasons

c) Line 172-176, how many groups stated ? This para seems more related to a communication skills rather than a direct result of poor relationship

d)Line 195 not sure if this is correct, "all groups but one " disclose problem of adherence ?

e) Line 244-248 showed the perception of AMR in farming, however, not sure what author tried to express here

3)Discussion

a)FG natural limitations did not discuss sufficiently

6. PLOS authors have the option to publish the peer review history of their article (what does this mean?). If published, this will include your full peer review and any attached files.

Reviewer #1: No

Reviewer #2: No

---

## [Author Response · Author response to Decision Letter 0]

12 Aug 2020

Dear Editor:

Thank you very much for allowing us to review our manuscript so that you appreciate its publication by PLoS One.

In the following lines we try to answer all your questions and comments.

You have been very kind to review our manuscript.

We have revised the style requirements and believe that the manuscript is now adapted.

- 2. Please include additional information regarding the interview guidelines used in the study and ensure that you have provided sufficient details that others could replicate the analyses.

The script followed for conducting and directing the focus groups is attached, to incorporate the manuscript as supplementary material

- 3. Please ensure that you include a title page within your main document. We do appreciate that you have a title page document uploaded as a separate file, however, as per our author guidelines (http://journals.plos.org/plosone/s/submission-guidelines#loc-title-page) we do require this to be part of the manuscript file itself and not uploaded separately.

A title page has been included at the beginning of the manuscript, as recommended. Thank you very much.

- 4. Please include captions for your Supporting Information files at the end of your manuscript, and update any in-text citations to match accordingly. Please see our Supporting Information guidelines for more information:

It has been revised, We consider that it is now correct. Thank you

Dear Reviewers, 

Thank you very much for allowing us to review our manuscript. We consider the topic of abuse and misuse of antibiotic, as you have said, a serius public heath concern. We wanted to make the most of our research. Thanks for appreciating it. We consider your suggestions have contributed positively to our article. Also, we have reviewed the language, thank you.

In its current version, our manuscript has won in quality, waiting now that meets the requirements of your magazine to be revalued and if necessary, be published in the future.

We have tried to respond to each of the contributions and suggestions that have been made to us get. 

Thank you. 

Reviewer #1: General comments:

The topic is of great interest where antibiotics abuse and misuse is a serious public health concern.The authors used adequate and solid methods for qualitative research.

Providing the scripts that the authors used to conduct the sessions as appendices facilitates the replication of the methods. Manuscript would benefit from a thorough proof read by a native English speaker to improve grammar.

Thank you for your apreciation. We consider the topic of abuse and misuse of antibiotic as you have said a serius public heath concern. We wanted to make the most of our research. Thank you. We have reviewed the language, thank you. We have included de script used to conduct the session in english and in spanish as you have suggest. Thank you.

Specific comments:

Line 43: spell out DDD when used for the first time in text

Thank you for notices this. We have already changed DDD for Define Daily dose. As you can see in the manuscrit: <<Most antibiotic use (80% to 90%) occurs in the outpatient setting, In terms of antibiotic consumption, Spain not only ranks highest among developed countries (in excess of 40 Define Daily Dose (DDDs) per 1,000 inhabitants per year).>>

Line 143: Better explain in the methods section the rationale for dividing FGs between over 65 and under 65.

Thank you for your sugestion. We have rewritten this paragraph in methods section. As you can read in page 3 line 76-89 of the document in highlight: <<We sought to ensure a high degree of heterogeneity in the composition of the groups in terms of age (over and under 65 years), urban or rural origin, and educational/professional qualifications, in order to cover the widest range of opinions (Table 1). We made groups following age criteria to explore the differences in knowledge and attitudes between retirees and workers, we also took into account the origin criteria due to possible differences in access to the health system. >>

Line 163: authors should explain what they mean by M1 (same for lines 168, 171, 183…)

Thank you for notices this. We have made a translate mistake, this is because in spanish the quote "M" is use for <<mujer>> that means woman, and H for <<hombre>> that means male. The quotes were typed incorrectly. Sorry for the misunderstanding, we have already changed this for the correct quotes. 

Line 176: authors should explain what they mean by H.1

Thank you for notices this. We have made a translate mistake, this is because in spanish the quote "H" is use for <<Hombre>> that means male, and M for <<Mujer>> that means woman. The quotes were typed incorrectly. Sorry for the misunderstanding, we have already changed this for the correct quotes. 

Lines 356-357: Conclusions are usually free of citations especially if the articles are cited earlier in text; unless they mention a quote or an expert opinion. 

Thank you for your consideration. We have included this change. 

Table 1: Providing more details about the baseline characteristics would give more insights to the respondents’ characteristics and would help understand the data better.

We have to apologize but we did not make that type of questions. In order to help the reader to understand data better we have included in table 1 in groups of <65 years a more specific rage of age being aware of the difference that could exist between participants under 35 years and over 50. We are not able to introduce more details of the baseline respondents characteristics because the only inclusion criteria were being older or younger of 65 age, this would clasificated the participant in one or other group. We have asked their work and fomartion to detect posible variations in the sessions. We also took into account the origin (urban or rural) criteria. 

Reviewer #2: This is an important topic and authors made a clear important argument on the study purpose. However, there are important parts which require significant revision, also language requires revision and polishing as it is difficult to understand.

1) Introduction:

a)1st para: unclear what author tried to express. Line 35--(delete "it has on"); Line 38-39 please revise and double check the english grammar

Thank you. We have made the changes you have suggested. Now in the manuscript you can read: <<Taken together, antibiotic adverse effects, ineffectiveness and resistance is one of the biggest threats to global health,i due to the great impact on morbidity, mortality and costs. Over- and misuse of antibiotics contributes significantly to this problem. Indeed, overuse must be assumed to account for the differences in antibiotic use (as much as threefold) among European Union countries, due there is no evidence of any difference in the prevalence of infectious diseases.>>

b)Line 51 change these drugs to antibiotics

Thank you, we have included that change. 

c)Line 57 abuse, can you define "abuse" and add this in the 2nd para

Thank you for yor question. After your suggestion we have changed abuse for misuse. Misues is the most used term in scientific researchs and the term that best expresses what we wanted to say. Thank you for your appreciation. 

2)Methods

a) Line 72, the references 18,19 did not match in Line 68-72

Thank you for your apreciation. We have changed this. 

b) Line 76-79, how the snowball method approved to be high degree of heterogeneity?

Thank you for your question. We have used snowball method using diferent Key informants. We made two categories of groups, taking into account age (over and under 65 years) to detect diferences in the knowledge and attitudes between retired and workers. Also, we asked if they were from the rural or urban due to the possible differences in the acces to the health care system. With this strategy we were able to made heteregeneus categories of groups but with homegeneus participants in each session. 

c) Line 85, how did the authors know the "saturation of information had been achieved" without knowing what these two groups' discussion

Thank for your question. After each FG we made a summary with the main ideas that came up during the sesision. Next step was to literaly transcript the sessions. After this, we made an indeep analysis of the trascription to identify the ideas as results by two individual researchers. In order to represent the saturation of the information we made table 3. In this table we have collected all the repeated ideas in each group. We have made 11 FG, when we realiced that no new ideas came up, after the analysis of each session, we stop doing new FG. 

d) Line 92: what was the response rates for requesting other authors

Thank you for your question. We have sent an email to all the authors of the published articles of qualitative methodology about this topic. We have send 10 email (concordant with references 24-33) and we reciebed 4 answers (concordant with references 28, 31-33). Also we used the scripts of the articles of general practicioners and pharmacist (concordant with reference 12 and 23 )

e) Ethical approval reference No ?

Thank you. We have included Ethical approval refenrece as you can read in line 114- 119. The study was evaluated and approved by the Santiago-Lugo Research Ethics Committee. After being informed of the purpose of the study and the fact that the sessions were to be recorded and transcribed but kept anonymous, all the participants agreed to take part and gave their written informed consent.

f) It was unclear if the discussion was conducted in Spanish or English transcript was translated into english ? how did the author make sure there was a back translation?

Thank you for your question. The discussion was conducted in Spanish or Galician (all the researchers who did the FG sessions were native speakers in both languages). After the sessions, the researchers (native speakers in both languages) did the literal transcription. Transcript analysis was also performed by native researchers. Finally, the researchers selected the sentences that most represented the results of the investigation, and then those were translated into Spanish (the ones that were in Galician Language). The next step was to send the entire manuscript to a native English speaker. The last step was to carefully read the results to make sure a good quality translation was done.

3) Results

a)Line 143 and Line 147 over-65? over 65 age, please double check the English grammar

Thank you for your suggestion. We have made thouse changes. 

b) add the definition of the summarised 5 main reasons

Thank you for your suggestion. We consider that table 2 define the 6 main reasons. As you had appreciated the used terms are not exactly the same in table 2 and main text, we have change this in the table for traing to being more precise. Thank you.

Descriptions of the main reason, as you can see in main text (table 2) are:

 (i) Lack of knowledge about antibiotics: Difficulties in differentiating antibiotics from other medications or consider that antibioctics are use for any infection

 (ii) Problems in the doctor-patient relationship: Lack of trust in physician (pressure on physician), consider that the physician supplies little information about the disease or consider time of consultation to be insufficient.

 (iii) Reason of problems of adherence (not finishing the entire treatment): Lack of credibility of professional judgement, improvement after initial doses, side effects of antibiotics, abandoning the treatment in order to be able to consume alcohol or oversights and carelessness 

 (iv) Use without prescription (using alternatives): trusted pharmacy, home medicine cabinet/leftover antibiotics or internet.

 (v) lack of perception of the problem of development of resistance: Do not think that there is any problem at present, excess use of antibiotics is not linked to advance of resistance or not considered to be a Public Health problem

 (vi) Internan Responsibility : inappropriate use of antibiotics considered responsible for the problem. External Responsability: considering other the responsable of the problem such as physicians, pharmaceutical industry, food, economic reasons, excess use in the past considered responsible for the problem. 

c) Line 172-176, how many groups stated ? This para seems more related to a communication skills rather than a direct result of poor relationship

We have change the sentence in order to be more exact. We understand that it could lead to misunderstanding. Thank you for your appreciation. In the manuscript now you can read:

<<Only two groups saw the medical practitioner as being responsible for taking the decision to prescribe antibiotics, once the necessary check-up and examination had been performed: [“I think it is necessary a severe control in the antibiotics. Doctors are the ones who always have to make the decision (taking or not antibiotics)”] (W6, FG1). Other groups stated that in some illness any person could be able to know that you need an antibiotic, even without a medical examination: [“Here with all the cold we have, you can get an urine infection. A simple urine infeccion and you don’t have more remedy than take an antibiotic.”] (W4, FG5).>>

d)Line 195 not sure if this is correct, "all groups but one " disclose problem of adherence ?

Thank you for your question. Of the 11 FG made, 10 stated problems in adherence due to lack of credibility of professional judmend, improvement after initial doses, side effects of antibiotics, abandoning the treatment in order to be able to consume alcohol or oversigths or carelessness. Only one group didnt declare any of those reason for stoping antibiotics before ending the treatment. 

e) Line 244-248 showed the perception of AMR in farming, however, not sure what author tried to express here

Thank you for your appreciation. We wanted to express that although the preocupaction about antibiotic resistance, general population do not identify the misuse of antibiotic as a main reason. General public point intensive farming as the guilty of the antibiotic resistance. We have changed this paragraph in order to be more clear with or intention. In main text now you can read: Although antibiotic resistance is of concern to the public, its advance is not associated with misuse. Difficulty in finding effective antibiotics is considered a consequence of intensive farming and food, insted of human misuse: [“All the chickens that come from intensive farming, for example, have antibiotics; and the cattle...have them in the meat as well as in the milk…”]

3)Discussion

a)FG natural limitations did not discuss sufficiently

Thank you for your apprecitation. We have included FG natural limitations. Now in the manuscript you can read: 

<<The FG sessions took place in Galicia, an area with a population that has a high use of antibiotics without prescription. Prudence is therefore called for when generalising the findings to other regions of Spain. It is necessary to replicate this workin other parts of Spain. Other natural limitation include the non-random sample, participants were volunteer. We don’t see this as a big limitation because the participants represented a wide range of ages, origin and formation.>>

---

## [Decision Letter · Decision Letter 1]

8 Oct 2020

PONE-D-20-05804R1

Factors determining antibiotic use in the general population: a qualitative study in Spain.

PLOS ONE

Dear Dr. Vazquez-Lago,

Thank you for submitting your manuscript to PLOS ONE. After careful consideration, we feel that it has merit but does not fully meet PLOS ONE’s publication criteria as it currently stands. Therefore, we invite you to submit a revised version of the manuscript that addresses the points raised during the review process.

We look forward to receiving your revised manuscript.

Kind regards,

Vijayaprakash Suppiah, PhD

Academic Editor

PLOS ONE

Reviewers' comments:

Reviewer's Responses to Questions

**Comments to the Author**

1. If the authors have adequately addressed your comments raised in a previous round of review and you feel that this manuscript is now acceptable for publication, you may indicate that here to bypass the “Comments to the Author” section, enter your conflict of interest statement in the “Confidential to Editor” section, and submit your "Accept" recommendation.

Reviewer #1: All comments have been addressed

Reviewer #2: (No Response)

2. Is the manuscript technically sound, and do the data support the conclusions?

Reviewer #1: Yes

Reviewer #2: Partly

3. Has the statistical analysis been performed appropriately and rigorously? 

Reviewer #1: N/A

Reviewer #2: Yes

4. Have the authors made all data underlying the findings in their manuscript fully available?

Reviewer #1: Yes

Reviewer #2: Yes

5. Is the manuscript presented in an intelligible fashion and written in standard English?

Reviewer #1: Yes

Reviewer #2: No

6. Review Comments to the Author

Reviewer #1: All comments addressed to the extent possible. The manuscript is technically sound, and the data support the conclusions. Recommend to accept manuscript for publication.

Reviewer #2: 1) The most important revision is to get a very careful english proofreading

2) In the abstract, the current conclusion is unclear and is not from the key findings.

3) The reasons why authors chose over and under 65 years old were not explained well.

4) Each quote shall include participant's age. (eg, 45yo W)

5) Add how the transcription and translation were conducted.

7. PLOS authors have the option to publish the peer review history of their article (what does this mean?). If published, this will include your full peer review and any attached files.

Reviewer #1: **Yes: **Elsy Ramia, PharmD, MPH, BCPS

Reviewer #2: No

---

## [Author Response · Author response to Decision Letter 1]

2 Dec 2020

Reviewer's Responses to Questions

Comments to the Author

1. If the authors have adequately addressed your comments raised in a previous round of review and you feel that this manuscript is now acceptable for publication, you may indicate that here to bypass the “Comments to the Author” section, enter your conflict of interest statement in the “Confidential to Editor” section, and submit your "Accept" recommendation.

Reviewer #1: All comments have been addressed

Reviewer #2: (No Response)

2. Is the manuscript technically sound, and do the data support the conclusions?

Reviewer #1: Yes

Reviewer #2: Partly

3. Has the statistical analysis been performed appropriately and rigorously?

Reviewer #1: N/A

Reviewer #2: Yes

4. Have the authors made all data underlying the findings in their manuscript fully available?

Reviewer #1: Yes

Reviewer #2: Yes

5. Is the manuscript presented in an intelligible fashion and written in standard English?

Reviewer #1: Yes

Reviewer #2: No

Thank you for your suggestion. We have submitted the article for review by a native English speaker________________________________________

6. Review Comments to the Author

Reviewer #1: All comments addressed to the extent possible. The manuscript is technically sound, and the data support the conclusions. Recommend to accept manuscript for publication.

Reviewer #2: 1) The most important revision is to get a very careful english proofreading

Thank you for your suggestion. We have submitted the article for review by a native English speaker

2) In the abstract, the current conclusion is unclear and is not from the key findings.

Thank you, We have changed the conclusions of the abstract in order to be more clear

3) The reasons why authors chose over and under 65 years old were not explained well.

Thank you, we have included a new explanation in page 3 line 80-83:

We decided to made this two groups to better explore the differences in the acces to the heathcare facilities (assuming more time in retirees), and also to explore the differences in the relationship with the doctor between older and yougers. 

4) Each quote shall include participant's age. (eg, 45yo W)

Thank you for your suggestion 

We have to apologize but we dont have this information available. We have the range of age of each group but we dont have the exact age of each participant. We have included the group age in order to make easier the reading. 

5) Add how the transcription and translation were conducted.

Thank you for your apreciation. We have included how transcription were conducted in page 4 line 109-112. The transcripts were first analyzed to obtain the results of the study. Subsequently, those that best represented the results were identified and selected. Only these quotes were sent for translation by a native English speaker. 

7. PLOS authors have the option to publish the peer review history of their article (what does this mean?). If published, this will include your full peer review and any attached files.

Do you want your identity to be public for this peer review? For information about this choice, including consent withdrawal, please see our Privacy Policy.

Reviewer #1: Yes: Elsy Ramia, PharmD, MPH, BCPS

Reviewer #2: No

---

## [Decision Letter · Decision Letter 2]

21 Jan 2021

Factors determining antibiotic use in the general population: a qualitative study in Spain.

PONE-D-20-05804R2

Dear Dr. Vazquez-Lago,

We’re pleased to inform you that your manuscript has been judged scientifically suitable for publication and will be formally accepted for publication once it meets all outstanding technical requirements.

Kind regards,

Vijayaprakash Suppiah, PhD

Academic Editor

PLOS ONE

Reviewers' comments:

Reviewer's Responses to Questions

**Comments to the Author**

1. If the authors have adequately addressed your comments raised in a previous round of review and you feel that this manuscript is now acceptable for publication, you may indicate that here to bypass the “Comments to the Author” section, enter your conflict of interest statement in the “Confidential to Editor” section, and submit your "Accept" recommendation.

Reviewer #1: All comments have been addressed

Reviewer #2: (No Response)

2. Is the manuscript technically sound, and do the data support the conclusions?

Reviewer #1: Yes

Reviewer #2: Yes

3. Has the statistical analysis been performed appropriately and rigorously? 

Reviewer #1: Yes

Reviewer #2: Yes

4. Have the authors made all data underlying the findings in their manuscript fully available?

Reviewer #1: Yes

Reviewer #2: Yes

5. Is the manuscript presented in an intelligible fashion and written in standard English?

Reviewer #1: Yes

Reviewer #2: No

6. Review Comments to the Author

Reviewer #1: All comments have been addressed, and all findings adequately presented. Manuscript can be accepted for publication.

Reviewer #2: Thanks for addressing the specific comments. A further language polishing or proofreading will be necessary to meet the journal standards.

7. PLOS authors have the option to publish the peer review history of their article (what does this mean?). If published, this will include your full peer review and any attached files.

Reviewer #1: **Yes: **Elsy Ramia, PharmD, MPH, BCPS

Reviewer #2: No

---

## [Editor Report · Acceptance letter]

25 Jan 2021

PONE-D-20-05804R2 

Factors determining antibiotic use in the general population: a qualitative study in Spain. 

Dear Dr. Vazquez-Lago:

I'm pleased to inform you that your manuscript has been deemed suitable for publication in PLOS ONE. Congratulations! Your manuscript is now with our production department. 

Kind regards, 

on behalf of

Dr. Vijayaprakash Suppiah 

Academic Editor

PLOS ONE